# Role of Lipids and Divalent Cations in Membrane Fusion Mediated by the Heptad Repeat Domain 1 of Mitofusin

**DOI:** 10.3390/biom13091341

**Published:** 2023-09-02

**Authors:** Anaïs Vlieghe, Kristina Niort, Hugo Fumat, Jean-Michel Guigner, Mickaël M. Cohen, David Tareste

**Affiliations:** 1Université Paris Cité, Institute of Psychiatry and Neuroscience of Paris (IPNP), Inserm UMR-S 1266, Team Membrane Traffic in Healthy & Diseased Brain, 75014 Paris, France; 2Sorbonne Université, Institut de Minéralogie, de Physique des Matériaux et de Cosmochimie (IMPMC), CNRS UMR 7590, MNHN, IRD UR 206, 75005 Paris, France; 3Sorbonne Université, Institut de Biologie Physico-Chimique (IBPC), CNRS UMR 8226, Laboratoire de Biologie Moléculaire et Cellulaire des Eucaryotes, 75005 Paris, France

**Keywords:** mitochondria, membrane, fusion, Mitofusin, amphipathic helix, divalent cations, lipid packing defects

## Abstract

Mitochondria are highly dynamic organelles that constantly undergo fusion and fission events to maintain their shape, distribution and cellular function. Mitofusin 1 and 2 proteins are two dynamin-like GTPases involved in the fusion of outer mitochondrial membranes (OMM). Mitofusins are anchored to the OMM through their transmembrane domain and possess two heptad repeat domains (HR1 and HR2) in addition to their N-terminal GTPase domain. The HR1 domain was found to induce fusion via its amphipathic helix, which interacts with the lipid bilayer structure. The lipid composition of mitochondrial membranes can also impact fusion. However, the precise mode of action of lipids in mitochondrial fusion is not fully understood. In this study, we examined the role of the mitochondrial lipids phosphatidylethanolamine (PE), cardiolipin (CL) and phosphatidic acid (PA) in membrane fusion induced by the HR1 domain, both in the presence and absence of divalent cations (Ca^2+^ or Mg^2+^). Our results showed that PE, as well as PA in the presence of Ca^2+^, effectively stimulated HR1-mediated fusion, while CL had a slight inhibitory effect. By considering the biophysical properties of these lipids in the absence or presence of divalent cations, we inferred that the interplay between divalent cations and specific cone-shaped lipids creates regions with packing defects in the membrane, which provides a favorable environment for the amphipathic helix of HR1 to bind to the membrane and initiate fusion.

## 1. Introduction

Mitochondria possess a double-membrane structure, consisting of the inner mitochondrial membrane (IMM) that encloses the matrix and the outer mitochondrial membrane (OMM) that separates the intermembrane space from the cell cytoplasm. These double-membrane organelles form a dynamic network that constantly changes shape through fusion and fission processes. Maintaining a delicate balance between these two processes, collectively known as mitochondrial dynamics, is crucial for mitochondrial function in cellular energy generation and overall cell health [1]. In mammalian cells, the elongation of mitochondria through fusion is orchestrated by the spatio-temporal coordination of two key transmembrane proteins from the dynamin superfamily: Mitofusins (Mfn1 and Mfn2) and OPA1, which mediate the fusion of the OMM and IMM, respectively [2]. Mitofusins are composed of an N-terminal GTPase domain followed by a first heptad repeat domain (HR1), a transmembrane domain (TMD) and a second C-terminal heptad repeat domain (HR2). These different domains are essential for Mitofusin function but their exact mode of action in mitochondrial fusion is still not fully understood [3,4,5,6].

Based on recent structural studies [7,8,9,10], the current working hypothesis for the molecular mechanisms of OMM fusion involves a *cis-* (i.e., within the same membrane) and *trans-* (i.e., across two different membranes) oligomerization of Mitofusin proteins. This initial event would lead to a long-distance (20–30 nm) membrane docking step involving the trans-dimerization of GTPase domains [8,11]. Through GTP hydrolysis, membrane bridging trans-Mitofusin complexes would then transit from an open to a closed conformation, bringing the OMM into close apposition. The HR2 domain could stabilize this short-distance (5–10 nm) membrane docking step by forming a homodimeric antiparallel coiled-coil complex [5,8,11].

Bringing membranes in close apposition is the first step in membrane fusion, but it is not enough on its own. For fusion to occur, the membrane structure must also be destabilized to facilitate the merging of the lipid bilayers. Two recent studies have emphasized the crucial role of amphipathic helices within the Mitofusin sequence for triggering OMM fusion [12,13]. One study found that the HR1 domain mediated liposome fusion in vitro and was essential for mitochondrial fusion in situ [12]. The fusion activity was attributed to a conserved amphipathic helix located at the C-terminus of HR1, suggesting that HR1 induces fusion by interacting with and perturbing the lipid bilayer structure. A similar fusion mechanism has been described for the C-terminal amphipathic tail of Atlastin 1 (Atl1), another dynamin-like transmembrane protein involved in endoplasmic reticulum (ER) membrane fusion [14,15,16]. Interestingly, another study demonstrated that when the TMD of Mfn1 was replaced with that of Atl1, the resulting chimeric protein localized to the ER membranes and was capable of mediating ER fusion [13]. Furthermore, an amphipathic helix identified between the TMD and the HR2 domain of Mfn1 could effectively replace the C-terminal amphipathic tail of Atl1 in both in vitro liposome fusion and in situ ER fusion. Thus, this study demonstrates that Mfn1 can mediate fusion on its own, i.e., even in the absence of mitochondrial co-factors, and suggests that Atl1 and Mfn1 share similar membrane fusion mechanisms. It remains to be determined whether Mfn2 can also intrinsically induce membrane fusion outside of a mitochondrial context.

The lipid composition of mitochondrial membranes also plays a crucial role in facilitating mitochondrial fusion. The successive stages leading to membrane fusion involve the formation of energy-demanding intermediate membrane structures with high curvatures [17]. Lipids that can relieve this energy stress by inducing favorable membrane bending therefore facilitate fusion. Mitochondrial membranes contain specific lipids, such as phosphatidylethanolamine (PE) and phosphatidic acid (PA), both of which possess a cone-shaped structure with a small headgroup area compared to the cross-sectional area of their hydrophobic chains. As a result, they can induce negative membrane curvatures when present in the outer leaflet of the lipid bilayers. Additionally, there is a unique lipid found in mitochondria known as cardiolipin (CL). When bound to divalent cations such as Ca^2+^ or Mg^2+^, CL can also adopt a conical shape. Thus, these three lipids have the potential to facilitate mitochondrial fusion events.

Indeed, a previous study found that high concentrations of CL were required for in vitro liposome fusion mediated by OPA1 [18]. CL also allows for the generation of PA at the OMM through its cleavage by the mitochondria-localized phospholipase D (MitoPLD) enzyme. The depletion of MitoPLD in mammalian cells due to RNA interference (RNAi) led to reduced mitochondrial fusion, suggesting that PA production at the OMM is important for fusion. Similarly, a reduction in the PE levels in mammalian cells due to RNAi silencing of the enzyme phosphatidylserine decarboxylase (Pisd), which converts phosphatidylserine (PS) into PE, resulted in mitochondrial fragmentation, indicating an inhibition of mitochondrial fusion [19]. Yeast cells lacking Psd1, the homolog of mammalian Pisd, also exhibited impaired mitochondrial fusion [20]. Additionally, the yeast cells that lacked both Psd1 and the CL synthase Crd1 displayed an even more fragmented mitochondrial network [21]. These studies demonstrated the critical importance of PE, CL and PA lipids in promoting efficient mitochondrial fusion. However, the specific molecular mechanisms by which these lipids facilitate mitochondrial fusion events remain to be established.

In this study, we investigate the role of PE, CL and PA lipids in membrane fusion mediated by the HR1 domain of Mitofusin. Specifically, our focus is on exploring the impact of PE, CL and PA headgroups while maintaining similar acyl chains. We chose to employ 18:1 acyl chains which, along with 16:0 acyl chains, constitute the most abundant acyl chains among the various phospholipid species and mammalian tissues [22]. We also examine the interplay between these lipids and the divalent cations Ca^2+^ and Mg^2+^ in facilitating membrane perturbation and fusion via the amphipathic helix of HR1.

## 2. Materials and Methods

### 2.1. Chemicals

N-2-hydroxyethylpiperazine-N′-2-ethanesulfonic acid (HEPES, OmniPur grade), potassium hydroxide solution 47% (KOH 47%, EMSURE grade for analysis), potassium chloride (KCl, OmniPur grade), calcium chloride dihydrate (CaCl_2_, OmniPur grade), magnesium chloride hexahydrate (MgCl_2_, OmniPur grade), glycerol (Molecular Biology grade), tris(2-carboxyethyl)phosphine hydrochloride (TCEP, ≥98% GC), n-octyl-β-D-glucopyranoside (β-OG, ≥98% GC), n-dodecyl β-D-maltoside (DDM, ULTROL grade) and sodium dithionite (Analytical grade) were purchased from Merck (Darmstadt, Germany). 5-(N-2,3-dihydroxypropylacetamido)-2,4,6-tri-iodo-N,N′-bis-(2,3-dihydroxypropyl)isophthalamide (Nycodenz, ≥98%) was purchased from Proteogenix (Schiltigheim, France).

All the aqueous solutions were prepared using 18.2 MΩ ultra-pure water and filtered using sterile 0.22 µm polyethersulfone (PES) membranes.

1-palmitoyl-2-oleoyl-sn-glycero-3-phosphocholine (PC), 1,2-dioleoyl-sn-glycero-3-phosphoethanolamine (PE), 1′,3′-bis[1,2-dioleoyl-sn-glycero-3-phospho]-glycerol (sodium salt) (CL), 1,2-dioleoyl-sn-glycero-3-phosphate (sodium salt) (PA), 1,2-dioleoyl-sn-glycero-3-phospho-L-serine-N-(7-nitro-2-1,3-benzoxadiazol-4-yl) (ammonium salt) (NBD PS), 1,2-dioleoyl-sn-glycero-3-phosphoethanolamine-N-(lissamine rhodamine B sulfonyl) (ammonium salt) (Rho PE), 1,2-dioleoyl-sn-glycero-3-phosphoethanolamine-N-[4-(p-maleimidophenyl)butyramide] (sodium salt) (MAL) and 1,2-dioleoyl-sn-glycero-3-[(N-(5-amino-1-carboxypentyl)iminodiacetic acid)succinyl] (nickel salt) (NTA-Ni) were purchased from Avanti Polar Lipids (Alabaster, AL, USA) as chloroform solutions.

### 2.2. Peptides

The heptad repeat domain 1 (HR1) of human Mitofusin 1 (Mfn1) was synthesized using Fmoc solid phase peptide synthesis and purified via a one-step reverse-phase HPLC process (conducted by Proteogenix, Schiltigheim, France, with a final purity exceeding 95%). The resulting sequence, denoted as Mfn1-HR1, spanned from T350 to L420 with C411S and C418S substitutions and included either a C-terminal Cys or His_6_ tag. A 1 mg lyophilized aliquot was put on ice and solubilized by slowly adding 1 mL of ice-cold buffer H (25 mM HEPES/KOH, pH 7.4; 150 mM KCl; 10% (*v*/*v*) Glycerol). In the case of Cys-tagged HR1, 0.25 mM of TCEP was included in the buffer. The peptide solution was subjected to 2 min of vortexing at room temperature, followed by a 20-s sonication on ice to eliminate any potential aggregates. The sonication process comprised two cycles, each consisting of 10 s of sonication at 10 W followed by a 10-s pause, utilizing the UP200St ultrasonic homogenizer from Hielscher (Teltow, Germany) equipped with a 2 mm sonotrode. The peptide solution was snap frozen using liquid nitrogen and stored at −80 °C in 50 µL aliquots.

### 2.3. Liposomes

The liposomes were generated using the detergent-assisted approach [23]. A total of 1.2 µmol of the desired lipid mix dissolved in chloroform was desiccated in a glass tube for 10 min using a gentle stream of argon followed by an additional 2-h period under vacuum. The desiccated lipid film was reconstituted in 400 µL of buffer H, supplemented with 1% (*w*/*v*) β-OG through 30 min of vigorous vortexing at room temperature. To lower the detergent concentration below the critical micellar concentration, 0.33% (*w*/*v*), and allow for liposome formation, 800 µL of buffer H was gradually added. The liposomes containing detergent were then subjected to overnight flow dialysis against 4 L of buffer H to ensure the complete detergent removal. The liposomes, with a final lipid concentration of 1 mM, were preserved on ice and shielded from light for a duration of 2–3 weeks.

### 2.4. Multi-Angle Dynamic Light Scattering (MADLS)

A total of 100 µL of liposomes diluted to 50 µM in buffer H were added to a low-volume quartz batch cuvette (ZEN2112, Malvern Panalytical, Worcestershire, UK). Their size distribution was assessed at 37 °C utilizing the Zetasizer Ultra Red instrument (Malvern Panalytical, Worcestershire, UK) in the Multi-Angle Dynamic Light Scattering (MADLS) mode. This mode captures correlation functions from three scattering angles: back scatter (173 degrees), side scatter (90 degrees) and forward scatter (13 degrees).

### 2.5. Cryogenic Transmission Electron Microscopy (cryo-TEM)

A total of 5 μL of liposomes, diluted to a concentration of 250 µM in buffer H, were dropped onto a Quantifoil grid (Quantifoil Micro Tools GmbH, Großlöbichau, Germany). After removing the excess liquid from the grid using a filter paper, the grid was immediately frozen in liquid ethane to create a thin vitreous ice layer. This rapid freezing process was accomplished using a custom-made mechanical cryo-plunger. The sample was then inserted into a Gatan 626 cryo-holder, cooled with liquid nitrogen and transferred to the microscope for observation at a low temperature of −180 °C. The images were captured using a LaB6 JEOL JEM2100 cryo-microscope (JEOL, Tokyo, Japan) operating at 200 kV. The microscope was equipped with an UltraScan 1000 2k × 2k CCD camera (Gatan, Pleasanton, CA, USA) and a JEOL low-dose system (Minimum Dose System, MDS, JEOL, Tokyo, Japan) to protect the thin ice film from irradiation both before and during imaging.

### 2.6. Liposome Co-Flotation Assay

To evaluate the anchorage of HR1 to the liposome membrane, 100 µL of HR1 at 50 µM were mixed with 100 µL of liposomes at 1 mM. This mixture was then allowed to incubate for 1 h at 37 °C with periodic gentle mixing (1 min at 300 rpm every 9 min). A 50 µL aliquot was withdrawn for use as the input control (unfloated fraction), while the remaining 150 µL was mixed with 150 µL of Nycodenz 80% (*w*/*v*) in buffer H. The resulting mixture was transferred to a centrifuge tube (0.8 mL, Open-Top Thinwall Ultra-Clear Tube, 5 × 41 mm, Beckman Coulter, Brea, CA, USA) and layered with 250 μL of Nycodenz 30% (*w*/*v*) in buffer H, followed by an additional 50 μL of buffer H. The layered gradient was subjected to centrifugation at 192,000× *g* for 4 h at 4 °C using a SW 55 Ti Swinging-Bucket rotor (Beckman Coulter, Brea, CA, USA). From the uppermost layer, 37.5 μL of liposomes was carefully collected (floated fraction). For the analysis, 12 µL of either the unfloated or floated fraction was mixed with 4 µL of the sample buffer (NuPAGE LDS Sample Buffer 4X, Invitrogen, Waltham, MA, USA) and separated using electrophoresis on a polyacrylamide gel (NuPAGE 4–12%, Bis-Tris, 1 mm, Mini Protein Gel, Invitrogen, Waltham, MA, USA). The gel was subsequently stained with Coomassie G-250 (PageBlue Protein Staining Solution, Thermo Scientific, Waltham, MA, USA) and imaged using the ChemiDoc Touch Imaging System (Bio-Rad, Hercules, CA, USA). To quantify the band intensities within the unfloated and floated fractions, the software ImageJ (https://imagej.nih.gov/ij/ accessed on 21 August 2023) was employed.

### 2.7. FRET-Based Lipid Mixing Assay

For each experimental condition, we prepared two distinct sub-groups of liposomes while maintaining identical lipid compositions. However, in one sub-group, termed the “fluorescent donor liposomes”, we introduced the fluorescence resonance energy transfer (FRET) pair of fluorescent lipids: NBD PS and Rho PE. These fluorescent lipids were added at a concentration of 1.5 mol% each at the expense of PC lipids. In a flat bottom 96-well white polystyrene plate (Thermo Scientific, Waltham, MA, USA), 27 µL of non-fluorescent acceptor liposomes at 1 mM were combined with 21 µL of buffer H. For the experiments involving the presence of divalent cations, 27 µL of non-fluorescent acceptor liposomes at 1 mM were combined with 19 µL of buffer H and 2 µL of CaCl_2_ or MgCl_2_ at 30 mM in buffer H. The plate was pre-warmed at 37 °C for 7 min. Subsequently, 6 µL of fluorescent donor liposomes at 500 µM were carefully added to one side of the well, while 6 µL of HR1 at 250 µM were added to the other side. To initiate the fusion reaction, the plate was gently shaken to ensure proper mixing of the three solutions. The process of lipid mixing was monitored by tracking the fluorescence dequenching of the NBD probes, which resulted from their dilution into the acceptor liposomes. The NBD fluorescence measurements were taken at 1-min intervals over a 90-min period (excitation at 460 nm, emission at 535 nm, cutoff at 530 nm) using the SpectraMax M5 microplate reader (Molecular Devices, San Jose, CA, USA), which was maintained at a constant temperature of 37 °C. After 90 min, we introduced 10 µL of a 2.5% (*w*/*v*) DDM solution to dissolve the liposomes. This step allowed us to measure the NBD fluorescence at infinite dilution, Fmax(NBD). To normalize the data, we applied the following equation, which provides the percentage increase in the NBD fluorescence at time t, %F(NBD, t).
%F(NBD, t) = [F(NBD, t) − Fmin(NBD)]/[Fmax(NBD) − Fmin(NBD)],(1)
where Fmin(NBD) represents the minimum recorded NBD fluorescence value during the experiment.

### 2.8. Sodium Dithionite Assay

To determine the ratio of hemifused to fully fused liposomes, we employed sodium dithionite to selectively quench the fluorescence of NBD PS lipids located in the outer leaflet of the liposomes. A solution of sodium dithionite at 100 mM in buffer H was freshly prepared, and 5 µL of this solution was mixed with 33.3 µL of fluorescent donor liposomes at 1 mM. The mixture was then incubated at 37 °C for 15 min. Subsequently, 28.3 µL of buffer H was introduced to the mixture to achieve a final liposome concentration of 500 µM. The FRET-based lipid mixing assay was then performed as described above. Since sodium dithionite’s activity is lost after 10 min at 37 °C, this ensured that the fluorescence signal from NBD PS lipids in the inner leaflets remained unquenched, even when DDM was added to solubilize the liposomes.

The percentage of liposomes undergoing hemifusion at time t, H(t), is expressed by the following equation.
H(t) = 100 × [F_T_ (t) − F_I_ (t)]/[F_T_ (t) − αF_I_(t)],(2)
where F_T_ and F_I_ represent the normalized fluorescence dequenching signals in the absence and presence of pre-treatment with sodium dithionite (indicating the total lipid mixing and inner leaflet lipid mixing, respectively) and α corresponds to the proportion of lipids residing in the inner leaflet of the liposomes.

## 3. Results

### 3.1. Exploring HR1-Mediated Fusion with Distinct Lipid Anchors: The Influence of Phosphatidylethanolamine

To investigate the mechanisms and efficiency of HR1-mediated fusion, we reconstituted HR1 into phosphatidylcholine (PC) liposomes using a lipid-anchorage strategy and evaluated its fusion activity using a FRET-based lipid mixing assay [23]. We first chose the maleimide lipid-anchorage method [24] (Figure 1a). HR1 was modified to contain a single terminal Cys residue at its C-terminus, which allowed for its specific coupling to the liposomes containing 18:1 MPB PE (MAL lipids). This coupling strategy ensured that the orientation of HR1 on the liposomes was consistent with its orientation on the mitochondrial membranes. HR1 was introduced at the start of the lipid mixing assay (t = 0), and its capacity to mediate liposome fusion was monitored over a duration of 90 min. Using this approach, we observed that HR1 induced robust lipid mixing (Figure 1b,c), which was in agreement with our previous work [12]. In our earlier study, we also noted fusion events between the liposomes bearing HR1 and the protein-free liposomes, suggesting that HR1 triggered fusion through membrane destabilization. Through a detailed bioinformatics analysis of the HR1 sequence, we identified a conserved amphipathic helix at its C-terminal end with the potential to bind and perturb the membrane structures. This finding was further supported by circular dichroism experiments, demonstrating an increase in the α-helical structure of HR1 upon its interaction with the liposome membranes. In this new study, we sought to explore the potential impact of HR1 on the membrane structural integrity by employing cryo-electron microscopy (cryo-EM). We found that after incubation with HR1, many liposomes—appearing to have undergone fusion based on their larger size—exhibited disrupted or disappearing membrane structures, suggesting bilayer structure perturbation caused by HR1 (Figure 1d). While we cannot exclude that the preparation of cryo-EM samples may have contributed to this phenomenon, it is worth noting that these structures were reminiscent of those identified in previous cryo-EM studies where the PC liposomes were treated with surfactants [25].

Specific lipids can also have a significant impact on the membrane structure and biophysical properties. The inner and outer mitochondrial membranes are enriched with a high proportion of the cone-shaped non-bilayer forming phosphatidylethanolamine (PE) lipid. In fact, the OMM contains approximately 30 mol% of PE lipids. We aimed to explore the effect of PE on HR1-mediated fusion. Surprisingly, when 30 mol% PE was introduced to the MAL-containing liposomes, HR1-mediated liposome fusion was completely abolished (Figure 1b,c). Through the analysis of the HR1 density on the surface of the liposomes using a liposome co-floatation assay, we discovered that HR1 was unable to anchor to the MAL-containing liposomes in the presence of PE (Figure 2a,b). This lack of anchorage can be attributed to the interaction between the amine group of PE and the Maleimide group [26], resulting in quenching the Maleimide group and preventing its interaction with the Cys residue at the C-terminus of HR1.

To overcome this limitation, an alternative lipid-anchorage strategy was employed. HR1 was modified to include a C-terminal His_6_ tag, enabling its chemical coupling to the liposomes functionalized with 18:1 DGS-NTA(Ni) (NTA-Ni lipids) (Figure 1a). In this system, HR1 retained the ability to induce liposome fusion, although to a lesser extent compared to the MAL lipid system (Figure 1b,c). This decrease in the fusion efficiency can be attributed to the lower reactivity of the NTA-Ni group compared to the MAL group, resulting in a lower surface density for the HR1 molecules on the liposomes (Figure 2a,b). The surface density of fusion proteins is, in fact, known to be critical for fusion efficiency [27]. Interestingly, the inclusion of 30 mol% PE in the NTA-Ni-containing liposomes resulted in a more than two-fold increase in HR1-mediated fusion (Figure 1b,c). Importantly, this effect was not due to a higher density of HR1 on the liposome surface in the presence of PE (Figure 2a,b). Since a high membrane curvature is known to activate fusion [28,29,30], we also used multi-angle dynamic light scattering to check whether the presence of PE in the liposome membrane affected their size. We observed that the size distribution of the liposomes remained unchanged in the presence of PE (Appendix A), which ruled out the possibility that fusion was activated by liposomes with high curvatures.

In the remainder of the manuscript, we exclusively used PC liposomes functionalized with NTA-Ni lipids as our model system. Specific lipids were added at the expense of PC to modify the liposome composition and examine the role of these lipids in HR1-mediated membrane fusion.

### 3.2. Cardiolipin Inhibits HR1-Mediated Fusion

Cardiolipin (CL) is an exclusive component of the mitochondrial membranes within cells. It is typically found at an average concentration of 5 mol% in the OMM [31]. However, its local concentration can reach up to 20 mol% at sites of contact between the outer and inner mitochondrial membranes [31] where fusion might take place [32]. Therefore, to investigate the impact of CL on HR1-mediated liposome fusion, we studied the effects of two concentrations of CL: 5 and 20 mol%. Surprisingly, the addition of CL at either concentration resulted in a slight, concentration-dependent inhibition of HR1-mediated liposome fusion (Figure 3a,b). This was unexpected, as CL was thought to be important for mitochondrial fusion [33]. Since HR1 had a pKa of 6.1, it carried a net negative charge in our working buffer at a pH of 7.4. Considering that CL is also negatively charged, we investigated whether the fusion inhibition was due to the inefficient coupling of HR1 to the liposome surface caused by electrostatic repulsion. The surface density of HR1 on the CL-containing liposomes was actually slightly higher than that observed on the PC liposomes (Appendix A), ruling out the absence of HR1 on the liposome membranes as the reason for the reduction in fusion.

Previous studies have demonstrated that both pure CL liposomes and liposomes composed of an equimolar mixture of PC and CL lipids can undergo fusion when exposed to high concentrations (9 mM or higher) of divalent Ca^2+^ or Mg^2+^ ions [34,35]. Here, we aimed to investigate the influence of these cations, when added at a physiological concentration of 1 mM, on HR1-mediated liposome fusion. We found that the addition of 1 mM Ca^2+^ or Mg^2+^ ions resulted in a 30–40% increase in HR1-mediated fusion for PC liposomes (Figure 3a,b). In contrast, these ions had no discernible effect on HR1-mediated fusion for the liposomes containing either 5 or 20 mol% CL (Figure 3a,b). Of note, an increase of 30–40% in HR1-mediated fusion, similar to that obtained using PC liposomes, was also observed for the liposomes containing 30 mol% PE in the presence of 1 mM Ca^2+^ or Mg^2+^ ions (Appendix A).

### 3.3. Phosphatidic Acid Enhances HR1-Mediated Fusion in the Presence of Calcium

Phosphatidic acid (PA) is an important lipid in mitochondria that is present at approx. 2 mol% on average on mitochondrial membranes [36,37]. PA is primarily synthesized in the ER and transported from the ER to the OMM via contact sites between these two organelles. Alternatively, PA can be generated on the OMM through cleavage of CL caused by the MitoPLD enzyme. An increase in the PA level at the surface of the mitochondria caused by the action of MitoPLD was found to promote the close apposition of mitochondrial membranes, which is a crucial step in the fusion process [38]. Since MitoPLD interacts directly with Mitofusin, it could create localized regions of high PA concentrations around the fusion sites, similar to what has been observed around the fission sites previously [39]. Considering that the specific concentration of PA at fusion sites is unknown, we chose to investigate the impact of incorporating 10 or 30 mol% of PA into the liposome membranes on HR1-mediated fusion. The selection of 30 mol% PA, despite seemingly high, allowed us to compare the effects of the two cone-shaped lipids, PA and PE, when used at the same concentration on HR1-mediated fusion. As in the case of CL, the addition of PA at either concentration led to a slight, concentration-dependent reduction in HR1-mediated liposome fusion (Figure 4a,b).

In previous works, it was found that liposomes composed of complex lipid compositions, including 5 mol% PA, exhibited protein-free fusion in the presence of high concentrations of Ca^2+^ (3 mM or higher) [40,41]. Here, we examined the effect of a physiological concentration of Ca^2+^ and Mg^2+^ (1 mM) on HR1-mediated fusion using liposomes with either 10 or 30 mol% PA. We observed a more than two-fold enhancement in HR1-mediated fusion when the liposomes containing 30 mol% PA were exposed to 1 mM Ca^2+^ (Figure 4a,b), which was comparable to the fusion levels observed with the liposomes containing 30 mol% PE (Figure 1b,c). Importantly, this effect was specific to Ca^2+^ ions and was not observed in the presence of Mg^2+^ ions. Furthermore, we did not observe any influence of these cations (Ca^2+^ or Mg^2+^) on HR1-mediated fusion of the liposomes containing 10 mol% PA (Figure 4a,b).

### 3.4. Phosphatidylethanolamine, but Not Phosphatidic Acid, Facilitates Hemifusion by HR1

During membrane fusion events, a distinct intermediate stage called a hemifused structure often occurs. This structure is characterized by the fusion of the outer leaflets of the lipid bilayers, while the inner leaflets and internal contents remain separated [42]. Our previous study using the MAL lipid system revealed that HR1 has the capacity to induce both hemifusion and complete fusion between PC liposomes. In the FRET-based lipid mixing assay, we observed that, at the end of the experiment, 60% of PC liposomes underwent complete fusion, while the remaining 40% were in a hemifused state [12]. In this study, we aimed to estimate the capacity of HR1 to induce hemifusion versus complete fusion of PC liposomes functionalized with NTA-Ni lipids and including or not PE or PA lipids, which were identified here as activators of HR1-mediated fusion. To distinguish hemifusion from complete fusion events in the FRET-based lipid mixing assay, we treated the fluorescent donor liposomes with a solution of sodium dithionite before initiating the fusion experiment [43] (Figure 5a). Sodium dithionite exclusively quenched the fluorescence signal of NBD PS lipids residing in the outer leaflet of liposomes. As a result, only complete fusion events involving the mixing of inner leaflets produced an increase in the fluorescence during the FRET-based lipid mixing assay. By comparing the lipid mixing curves obtained using the liposomes treated or not treated with sodium dithionite, we were able to quantify the hemifusion events between the liposomes.

We observed that, at the end of the FRET-based lipid mixing assay, 40–50% of the PC liposomes functionalized with NTA-Ni lipids underwent hemifusion in the presence of HR1 (Figure 5b), similar to what we observed previously with the MAL lipid system [12]. Interestingly, the incorporation of 30 mol% PE in the liposome membrane promoted hemifusion events, with approx. 80% of the liposomes existing in a hemifused state at the end of the FRET-based lipid mixing assay (Figure 5b). The introduction of 1 mM Ca^2+^ or Mg^2+^ did not have any additional impact on the occurrence of hemifusion. Conversely, while the presence of PA and Ca^2+^ led to a significant enhancement of HR1-mediated fusion, it did not result in a corresponding increase in hemifusion events, as observed with PE (Figure 5b). This indicated that the activation of fusion by PE, with or without cations (Ca^2+^ or Mg^2+^) and by PA in the presence of Ca^2+^, likely involved partially distinct molecular mechanisms.

## 4. Discussion

In a previous study, we identified an amphipathic helix at the C-terminal end of the HR1 domain. We proposed that this amphipathic helix induced HR1-mediated fusion by perturbing the lipid bilayer structure, especially in membrane regions presenting lipid packing defects, caused by either a high membrane curvature or the presence of lipids with a cone-shaped structure [12]. In this study, through cryo-EM observations, we noticed that pure PC liposomes displayed locally disappearing membrane structures after incubation with HR1 (Figure 1d). This finding was consistent with a perturbation of the lipid bilayer structure when HR1 interacts with the liposome membrane.

The primary objective of this study was to examine the influence of the membrane lipid composition, both in the presence and absence of divalent cations (Ca^2+^ or Mg^2+^), on HR1-mediated liposome fusion. In the absence of cations, the inclusion of 30 mol% PE lipids in the liposome membrane resulted in a more than two-fold increase in the extent of lipid mixing mediated by HR1 compared to the liposomes composed of pure PC lipids (Figure 1b,c). We attributed this effect to the cone-shaped structure of PE, which induced packing defects in the membrane structure and, consequently, enhanced membrane binding and perturbation caused by the amphipathic helix of HR1. The presence of PE in the liposome membrane also promoted hemifusion events over full fusion mediated by HR1. The percentage of liposomes undergoing hemifusion increased from 50 to 80% when the liposomes contained 30 mol% PE lipids (Figure 5b). This effect can also be associated with the cone-shaped structure of PE, which is known to induce the transition from a bilayer to a non-bilayer inverted hexagonal phase structure with high local curvatures [44], a process believed to occur during the formation of the stalk/hemifused fusion intermediate [45].

Unlike PE, the inclusion of the cone-shaped PA lipid at either 10 or 30 mol% in the liposome membrane did not activate fusion by HR1 in comparison to pure PC liposomes. Instead, it led to a slight concentration-dependent inhibition (Figure 4a,b). This can be explained by the negative charge of the PA headgroup, which hindered the binding of HR1—carrying a net negative charge at pH 7.4—with the membrane. Similarly, the reduction in HR1-mediated fusion when the liposomes contained CL lipids can be attributed to the electrostatic repulsion between HR1 and the negatively charged CL lipids (Figure 3a,b). Furthermore, the presence of PA in the liposome membrane did not increase the number of hemifusion events caused by HR1 compared to pure PC liposomes (Figure 5b). This effect was likely due to electrostatic repulsions between the PA headgroups, preventing the clustering of PA lipids into domains of high local curvatures.

In the presence of Ca^2+^, HR1-mediated lipid mixing of the liposomes containing 30 mol% PA was strongly activated. Specifically, we observed a three-fold increase compared to the liposomes with 30 mol% PA in the absence of Ca^2+^ and a two-fold increase compared to pure PC liposomes in the presence of Ca^2+^ (Figure 4a,b). This activation can be attributed to the formation of PA-rich membrane domains facilitated by the presence of Ca^2+^, which allows for the attraction of PA headgroups [46]. At the boundary of these membrane domains, the hydrophobic chains of lipids are exposed, enabling the interaction of the amphipathic helix of HR1 with the membrane structure. This interaction can be further reinforced by electrostatic attractions between the negatively charged PA headgroup and clusters of positive residues within the HR1 sequence. The amphipathic helix of HR1 notably includes a cluster of three consecutive positively charged Lysine residues (395-KKK-397). Similar recognition motifs have been identified in PA-binding proteins, including those with an amphipathic helix [47,48,49]. As observed in the absence of Ca^2+^, hemifusion events were not promoted for the liposomes containing 30 mol% PA compared to PC liposomes, suggesting that zones of high curvatures enriched in PA did not form, even when the PA–PA headgroup attraction was facilitated by the presence of Ca^2+^. Surprisingly, in the presence of Mg^2+^ ions, the HR1-mediated fusion of the liposomes with 30 mol% PA was not activated. This was unexpected because Ca^2+^ and Mg^2+^ ions were found to have a similar capacity to induce the protein-free fusion of pure PA liposomes [50,51]. However, Mg^2+^ ions are known to be more hydrated than Ca^2+^ ions [52], which may limit their accessibility to PA headgroups that are buried deep below the PC headgroups in mixed PC:PA bilayers [47,53]. Such an accessibility issue would not occur in the protein-free fusion of pure PA bilayers. It would also explain why Ca^2+^ and Mg^2+^ have comparable effects on the HR1-mediated fusion of PC and PC:PE bilayers in our study (Appendix A), since the PC and PE headgroups were fully accessible. In these cases, the activation of fusion can be explained by the presence of a small fraction of negative charges in the PC bilayers [54], which may be sufficient to induce their interaction with divalent cations and promote the formation of packing defects facilitating HR1 membrane interaction.

In conclusion, we found that the interplay of divalent cations and specific cone-shaped lipids allowed for the formation of membrane regions with molecular packing defects, which, in turn, facilitated membrane perturbation and fusion mediated by the amphipathic helix of HR1. Future studies will need to address the role of lipids and divalent cations in the mode of action of the other functional domains of Mitofusin and their impact within the context of the full-length protein. It will also be important to consider the contribution of ER–mitochondria contact sites for regulating mitochondrial fusion events [55,56]. These sites, known for facilitating the lipid and Ca^2+^ exchange between the two organelles [57], may act as hotspots, leading to local increases in the concentration of fusogenic lipids and Ca^2+^ ions, thus promoting efficient Mitofusin-mediated fusion.

## Figures and Tables

**Figure 1 biomolecules-13-01341-f001:**
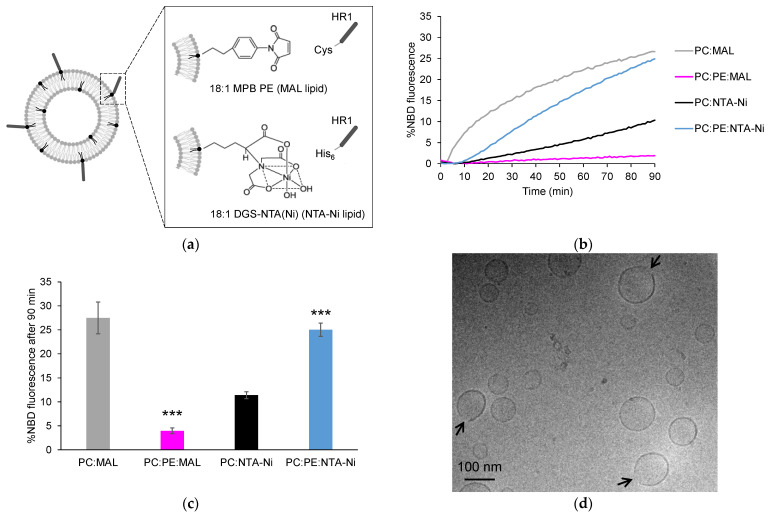
Influence of lipid anchors and PE on HR1-mediated fusion. (**a**) The peptides used in this study were HR1 fragments of Mfn1 with a Cys or a His_6_ tag at their C-terminus, allowing for their chemical coupling to the liposomes functionalized with either 18:1 MPB PE (MAL lipids) or 18:1 DGS-NTA(Ni) (NTA-Ni lipids), respectively. (**b**) Representative kinetics of a FRET-based lipid mixing assay between the liposomes containing 5 mol% of either MAL or NTA-Ni lipids in their membrane, along with 95 mol% PC or 65 mol% PC and 30 mol% PE. The fusion reaction was initiated by adding HR1-Cys or HR1-His_6_ peptides at t = 0, with 25 µM HR1 and 500 µM lipids in the reaction mix. The control experiments in which the buffer alone was added in place of HR1 are presented in Appendix A. (**c**) Average extent of lipid mixing observed after a 90-min period, based on the data from the n = 7 to 21 independent kinetics experiments, similar to the one presented in panel (**b**). The error bars represent the standard errors of the mean. Statistical comparisons were performed using two-sample *t*-tests against the condition without PE (*** *p* < 0.001). (**d**) Cryo-EM picture of the liposomes composed of 95 mol% PC and 5 mol% MAL after a 1-h incubation at 37 °C with HR1-Cys peptides (12.5 µM peptides and 500 µM lipids).

**Figure 2 biomolecules-13-01341-f002:**
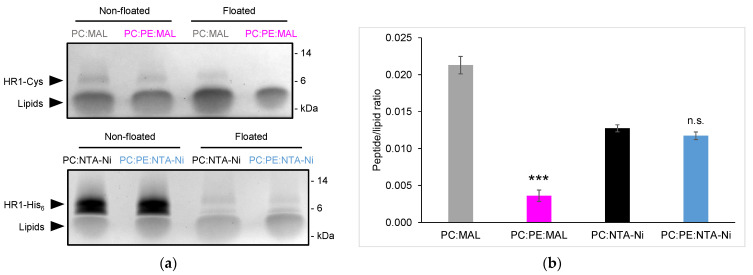
Surface density of HR1 on the liposome membrane. (**a**) Liposomes with the same lipid compositions as shown in Figure 1 were incubated with HR1-Cys or HR1-His_6_ peptides (500 µM lipids and 25 µM peptides) at 37 °C for 1 h (same color code as in Figure 1). The reaction mixes were separated using a discontinuous nycodenz gradient to distinguish the HR1-bound liposomes from unbound HR1. The protein and lipid recoveries in the floated samples were estimated using SDS-PAGE stained with Coomassie upon comparison with the non-floated samples. (**b**) The HR1-to-lipid ratios in the liposome membrane were estimated from the n = 3 to 5 independent experiments. The error bars represent the standard errors of the mean. Statistical comparisons were performed using two-sample *t*-tests against the condition without PE (n.s. *p* > 0.05; *** *p* < 0.001).

**Figure 3 biomolecules-13-01341-f003:**
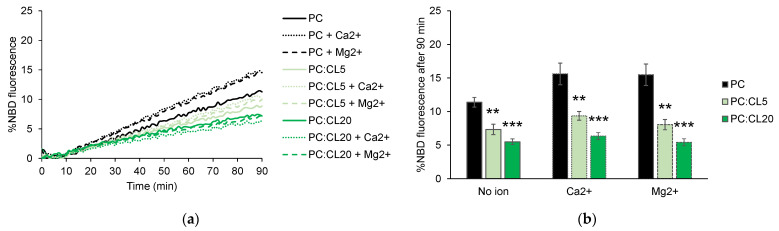
Effect of CL and divalent cations on HR1-mediated fusion. (**a**) Representative kinetics of a FRET-based lipid mixing assay between the liposomes containing 5 mol% NTA-Ni lipids in their membrane, along with different lipid compositions: 95 mol% PC (black), 90 mol% PC and 5 mol% CL (light green), or 75 mol% PC and 20 mol% CL (dark green). The fusion reaction was initiated by adding HR1-His_6_ peptides at t = 0 in the absence or presence of the divalent cations Ca^2+^ or Mg^2+^ (500 µM lipids, 25 µM peptides and 1 mM cations). The control experiments with the buffer alone instead of HR1 are presented in Appendix A. (**b**) Average extent of lipid mixing observed after a 90-min period, based on the data from the n = 3 to 21 independent kinetics experiments, similar to the one presented in panel (**a**). The error bars represent the standard errors of the mean. Statistical comparisons were performed using two-sample *t*-tests against the condition without CL and with the same ionic composition (** *p* < 0.01; *** *p* < 0.001).

**Figure 4 biomolecules-13-01341-f004:**
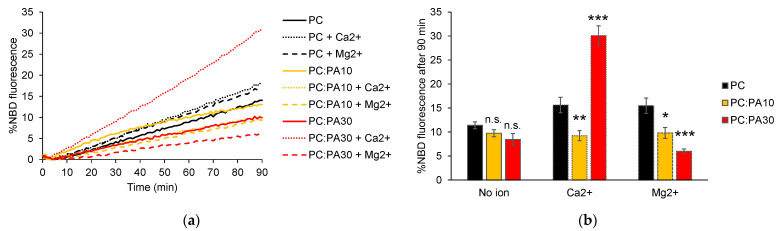
Effect of PA and divalent cations on HR1-mediated fusion. (**a**) Representative kinetics of a FRET-based lipid mixing assay between liposomes containing 5 mol% NTA-Ni lipids in their membrane, along with different lipid compositions: 95 mol% PC (black), 85 mol% PC and 10 mol% PA (yellow), or 65 mol% PC and 30 mol% PA (red). The fusion reaction was initiated by adding HR1-His_6_ peptides at t = 0 in the absence or presence of the divalent cations Ca^2+^ or Mg^2+^ (500 µM lipids, 25 µM peptides and 1 mM cations). The control experiments with the buffer alone instead of HR1 are presented in Appendix A. (**b**) Average extent of lipid mixing observed after a 90-min period, based on the data from the n = 7 to 21 independent kinetics experiments, similar to the one presented in panel (**a**). The error bars represent the standard errors of the mean. Statistical comparisons were performed using two-sample *t*-tests against the condition without PA and with the same ionic composition (n.s. *p* > 0.05; * *p* < 0.05; ** *p* < 0.01; *** *p* < 0.001).

**Figure 5 biomolecules-13-01341-f005:**
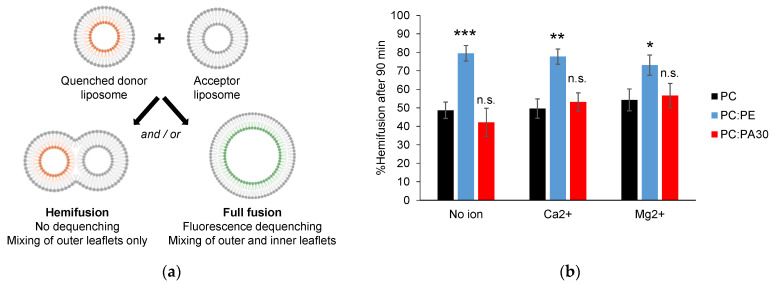
Effect of PE and PA on HR1-mediated hemifusion. (**a**) Hemifusion events were quantified using the sodium dithionite assay. Fluorescent donor liposomes were pre-treated with sodium dithionite to eliminate the NBD fluorescence of their outer leaflet, allowing for only full fusion events to lead to fluorescence dequenching in the FRET-based lipid mixing assay. (**b**) The percentage of liposomes that underwent hemifusion after 90 min was determined by comparing the fluorescence dequenching signals obtained with or without prior sodium dithionite treatment (n = 3 to 11 independent experiments; error bars represent standard errors of the mean). Statistical comparisons were performed using two-sample *t*-tests against the condition without PE or PA and with the same ionic composition (n.s. *p* > 0.05; * *p* < 0.05; ** *p* < 0.01; *** *p* < 0.001).

## Data Availability

The data is contained within the article or the Appendix A.

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
