# Peer review of "Role of Lipids and Divalent Cations in Membrane Fusion Mediated by the Heptad Repeat Domain 1 of Mitofusin"

_biomolecules, 2023, doi:10.3390/biom13091341_

Round 1
Reviewer 1 Report
The article presented by Anaïs Vlieghe et al. deals with a very interesting topic on the search for clues to understand the mechanism of action of the fusion proteins mitofusins. This study could serve to search for the keys involved in the process. Although the study is very well focused, there are many details missing that are necessary to validate the claims. Similarly, the assertions made in the article should have more robust experiments and more replications to back up the author's claims, which in the current state of the article are oversimplified and weak.
Although in its current form I do not recommend its publication for a journal as prestigious as Biomolecules. I would venture to comment on several points that would reinforce the impact of this study, which I list below:
Major comments:
1) It would be really interesting to have kinetic functions of HR1 peptide incorporation into vesicles. Have you not considered such a study? Why did you use these incubation times and not others in your protocol?
2) In relation to how the graphs are presented I have several comments.
All of them that talk about NDB fluorescence vs. time contain lines without any margin of error. How do we know that, for example, in figure 3a the observed differences are not within the margin of error? Have they used N=1 and therefore cannot include margin of error in those measurements?
The graphs containing margins of error and where there are differences have not been subjected to any statistical test. I do not know if there are significant differences and of what order. It would also be good to use all the points that make up these column diagrams.
In figure S1 I am unable to see any difference as the graphs have been plotted. Why have they been presented like this, with a y-axis that goes up to 35 in all cases and not shortened to a value of 15 (for example) in order to be able to observe and compare graphs? It is incomprehensible to me how this figure is represented.
In figure S2, concerning the size distribution, I do not understand why the logarithmic scale has been used. It makes it very difficult to see differences between vesicle sizes with different compositions. Is this form of representation standard or has it been seen elsewhere?
3) I have concerns about the conclusions regarding cardiolipin. Cardiolipin is a very diverse type of lipid: it is a family that contains a large number of variations in its aliphatic tails. For example, the cardiolipin present in bacteria has little to do with that present in most mitochondria. They behave very differently, as their chains have very marked alterations. Normally in this type of study I have seen reference to the use of cardiolipin from bovine heart, but here 18:1 is used. Why has this choice been made? Could you carry out the experiments with a cardiolipin like that from bovine heart to see if the conclusions are the same?
Regarding the conclusions, I would like to know if you could include answers to the following questions raised here:
1) If MFN1 by itself can promote fusion, what is the role of MFN2?
2) If the fragment containing HR1 (used here as a peptide) is capable of causing fusion, what are the other domains of the MFN1 protein used for?
In general, it is a well-constructed and well-argued article, but it lacks some of the key points made here. This would improve its robustness.
Author Response
Please find in the attachment a file that includes our point-by-point answers to the 2 reviewers’ feedback.

Reviewer 2 Report
In this current study, Vlieghe et al. discovered that the interaction between divalent cations and certain cone-shaped lipids enables the creation of membrane areas with molecular packaging defects. Consequently, this promotes membrane disturbance and fusion, which is mediated by the amphipathic helix of HR1 of mitofusin. This study holds significant potential interest for researchers engaged in the field of membrane fusion. However, a few minor concerns need to be resolved before it can be published in Biomolecules.
*Minor issues to be addressed:
1. In Figure 1D, the authors argue that some liposomes displayed locally disappearing membrane structures. Is this thermo-dynamically possible? Isn't this simply an experimental artifact that arises during the preparation of EM samples?
2. If HR1-mediated fusion occurs mainly through perturbation of the lipid bilayer structure, is it possible that a majority of liposomes undergo rupture and resealing, which also generates lipid-mixing FRET signals?
3. Using sodium dithionite, the authors found that PE arrests a majority of liposomes at the hemifusion state. Can the authors prove this using Cryo-EM? If 50~80% of liposomes are arrested at the hemifusion state, the authors should be able to observe a lot of docked liposomes after the fusion reaction.
Author Response

(The authors gave the same response as above.)

Round 2
Reviewer 1 Report
Dear authors,
Thanks for the point to point response with my concernings. However, I still find neccesary to perform the experiments suggested on my previous report. Any of the new experiments suggested have been taken into account so I cannot accept the manuscript in its actual form.
CL, which the key lipid on the membranes containing MFNs should be carefully taken into account. Bovine heart tissue, in which there are an excess of mitocondria (more energy needed) are one of the best examples of what kind of cardiolipin should be taken into account for experiments involving mitocondria proteins. Again, I should suggest to change CL lipid to one with the appropiate acyl chains.
Another important fact is the conclusions regarding "representative" graphs. For the scientific community, graphs with N=1 are not very useful, as they have a proper margin of error that must appear within the kinetic graph to compare with the rest of the situations. For me, if you already have the data, creating a graph regarding errors must be done / including.
For me, those concernings (beside others) are a must for the manuscript to be accepted (even more if already have been pointed out on the previous round).